# Status and Challenges of Qinghai–Tibet Plateau's Grasslands: An Analysis of Causes, Mitigation Measures, and Way Forward

**Moses Fayiah** [1,2] , **ShiKui Dong** [1,3,*], **Sphiwe Wezzie Khomera** [4], **Syed Aziz Ur Rehman** [5] , **Mingyue Yang** [1] **and Jiannan Xiao** [1]

1   State Key Laboratory of Water Environment Simulation, School of Environment, Beijing Normal University, Beijing 100875, China; moses.fayiah@yahoo.co.uk (M.F.); yueyuemuseum@163.com (M.Y.); xiaojiannan168@163.com (J.X.)
2   Department of Forestry, School of Natural Resources Management, Njala University, Njala 232, Sierra Leone
3   Department of Natural Resources, Cornell University, Ithaca, NY 14853-3001, USA
4   Department of Comparative Education, Faculty of Education, Beijing Normal University, Beijing 100875, China; swkhomera@gmail.com
5   Department of Environmental Sciences, University of Veterinary and Animal Sciences, Lahore 54000, Pakistan; syed.aziz@uvas.edu.pk
*   Correspondence: dsk03037@bnu.edu.cn or dongshikui@sina.com

**Abstract:** Grassland ecosystems on the Qinghai–Tibet Plateau (QTP) provide numerous ecosystem services and functions to both local communities and the populations living downstream through the provision of water, habitat, food, herbal medicines, and shelter. This review examined the current ecological status, degradation causes, and impacts of the various grassland degradation mitigation measures employed and their effects on grassland health and growth in the QTP. Our findings revealed that QTP grasslands are continually being degraded as a result of complex biotic and abiotic drivers and processes. The biotic and abiotic actions have resulted in soil erosion, plant biomass loss, soil organic carbon loss, a reduction in grazing and carrying capacity, the emergence of pioneer plant species, loss of soil nutrients, and an increase in soil pH. A combination of factors such as overgrazing, land-use changes, invasive species encroachment, mining activities, rodent burrowing activities, road and dam constructions, tourism, migration, urbanization, and climate change have caused the degradation of grasslands on the QTP. A conceptual framework on the way forward in tackling grassland degradation on the QTP is presented together with other appropriate measures needed to amicably combat grassland degradation on the QTP. It is recommended that a comprehensive and detailed survey be carried out across the QTP to determine the percentage of degraded grasslands and hence, support a sound policy intervention.

**Keywords:** degradation; biodiversity; grassland; grazing; soil; abiotic; climate change; China; biotic

## 1. Introduction

The Qinghai–Tibet Plateau (QTP) lies in the southwest of China and encompasses the entire Tibet Autonomous Region and Qinghai province, and parts of Yunnan, Sichuan, and Gansu provinces [1]. Scholarly ecologists like Dong et al. [2] and Liu et al. [3] referred to it as the "roof of the world", "center of species formation and differentiation [4], "third pole" [5], and "hot island" [4]. The QTP spreads roughly 1500 km north and 3000 km south with a total land area of approximately $2.5 \times 10^6$ km$^2$ [6,7], accounting for about 25% of the whole of China's territory [4,6]. The warmest month in QTP is July with a mean temperature fluctuating from 7 °C to 15 °C, while January remains the coldest

month having a temperature range from 1 °C to 7 °C. The QTP's average annual temperature is 1.6 °C, the precipitation is 413.6 mm per year [7–9], and 60%–90% of this precipitation falls during the wet humid summers (June–September), while 10% of the precipitation falls during the arid winters (November–February) [10]. The atmospheric pressure and density of the QTP range from 50–60% and 60–70%, respectively [11]. On the QTP, the most noticeable land cover changes are grassland degradation, deforestation, desertification, and permafrost decline [12]. There are $1.33 \times 10^8$ ha grasslands, representing 60% of QTP and 30% of China's grassland [1,13]. The QTP is considered as the highest plateau in the world [14], with an average elevation of 4000 m [15,16]. The grasslands of QTP are divided into seventeen types (Table 1), with alpine meadow being the highest (44.64%), followed by alpine steppe (28.75%), while eight grasslands are considered to be minor as each of them makes up less than 1% of total grasslands in China [7,17].

**Table 1.** Grassland type on the Qinghai–Tibet Plateau (adapted from Li et al. [7]).

| Type | Area (ha) | Rangeland (%) | Ecosystem | Distribution |
|---|---|---|---|---|
| Temperate meadow steppe | 3833 | 2.9 | | |
| Alpine meadow steppe | 5626 | 4.3 | | |
| Alpine steppe | 37,762 | 28.8 | | |
| Alpine desert steppe | 8679 | 6.6 | | |
| Temperate desert | 2084 | 1.6 | 58.8% | 23.0% |
| Alpine desert | 5967 | 4.9 | | |
| Temperate mountain meadow | 6067 | 4.6 | | |
| Alpine meadow | 58,652 | 44.6 | | |
| Total | 131,322 | 100.1 | | |

The QTP is characterized by extreme weather and environmental conditions and is one of the unique world habitats in terms of species biodiversity. The QTP gained importance as a biodiversity hotspot globally over a century ago [3]. The plateau displays numerous ecological functions and services, such as climate control, carbon storage, tourism and aesthetical recreation, water resources control, pastoral production, and more at both the regional and local scale [2,18]. Other ecosystem services provided by the plateau include, for example, carbon sequestration, fiber production, conservation, biomass productivity, and recreation [18]. The plateau serves as a headwater station for Asia's largest rivers [5], is least polluted, and is rich in solar energy, hydro-energy, geothermal energy, and mineral resources [19]. Its vast grassland location, speciation, climate, altitude, and unique biogeographic landscape [4] makes the plateau a world ecological reservoir of alpine biodiversity. Furthermore, the QTP has been found to host the world's largest ecosystem of alpine grasslands. The alpine grasslands take up the key ecosystems of QTP, serving as an eco-safety barrier for animal husbandry in highlands [19]. The QTP plays essential roles in climate change regulation or drive, and is considered to be an amplifier of world climate change [5,20], hosting approximately 80 nature reserves mostly in the southeastern region of China [4].

In terms of biodiversity, the QTP hosts 210 mammal species, 12,000 plant species, 532 bird species, and 5000 epiphyte species [4]. The plateau has approximately 3760 unique spermatophyte species, 300 rare and endangered species of high plants, 120 species of rare and endangered animals, and 280 unique vertebrate species. According to Li et al. [3] and Anon, [19], the plateau has 17 out of the total 18 types of grasslands in China and hosts two biological territories, namely, the Holarctic and Paleotropical kingdoms [4]. The QTP is famous for a variety of rare and endemic species, and it harbors more than 1000 species of medicinal importance as well as many germplasm resources [4]. The plateau is a national treasure for biodiversity in China and even the world at large.

Across the QTP, academic publications covering diverse scope on the plateau have given insights on the ecological status of the alpine grasslands in recent times. A recent study by Wu et al. [21] assessed the impacts of grazing exclusion on productivity partitioning along with regional plant diversity and climatic gradient on QTP's alpine grasslands. Yu et al. [22] assessed the soil quality under

different land uses on alpine grasslands using the soil quality index. Liao et al. [23] demonstrated how ecological restoration enhanced the ecosystem health using the pressure state response (PSR). Neuenkamp et al. [24] reported the benefits of mycorrhizal inoculation (MI) for ecological restoration using a meta-analysis; and Liu et al. [25] explored the spatial and temporal degradation levels of grasslands in the three-rivers headwater region. Huang et al. [26] determined the possibilities of ecological conservation and restoration through payment for ecosystem services within the QTP.

Additionally, numerous scholars have conducted studies ranging from vegetation restoration [27], plant diversity [28], grassland biomass productivity, soil properties and characteristics, and grazing impacts on the ecology [7] to soil quality assessment on the alpine grassland ecosystems of the QTP [29]. Their contributions have provided a solid literature basis for subsequent studies within the QTP region. These studies have further narrowed the knowledge gap in land degradation and causes as well as the mitigation measures taken over the years. Grassland degradation, however, is threatening the sustainability of the QTP. A study by Wen et al. [29] estimated that economic loss due to land degradation was $198/ha in 2008.

Tackling grassland degradation is a global challenge, especially with climate change in the equation [30]. Understanding the drivers of grassland degradation on the Qinghai–Tibet Plateau and the appropriate mitigation measures to combat the grave situation is very critical for future sustainability. The knowledge of critical drivers of degradation and their root causes can help in adopting appropriate restoration methods and strategies to mitigate the degradation of the grassland in perpetuity. Additionally, adequate grassland degradation information is critical for appropriate restoration and conservation of vegetation across the grasslands [31]. According to Fassnacht et al. [32], the bone of contention about grassland degradation on the QTP is a poor understanding of the degradation processes as a result of inadequate information on the socioeconomic and ecological baseline and the absence of monitoring on a long-term basis [29,32]. Therefore, all types of grassland management interventions have to be developed to adapt to the climate change implications [2,33]. Due to the changing dynamics of the alpine grassland ecosystem stimulated by climate change and human disturbances, there is still a knowledge gap to be filled within this scope.

This article is tailored to review the current ecological status and challenges on Qinghai–Tibet Plateau's grasslands and analyze the causes, mitigation measures, and way forward. We hypothesized in this review: (1) climate change and human activities have changed the ecological status of the Qinghai–Tibet Plateau and (2) adequate mitigation/restoration measures can help to maintain the biodiversity of the grasslands. The review is aimed at assessing the status and challenges of Qinghai–Tibet Plateau's grasslands by analyzing the degradation drivers, countermeasures employed, and the sustainable way forward in conserving the biodiversity of the plateau.

Furthermore, the review concludes and recommends some of the best grassland management practices to ensure the effective restoration of the grassland ecosystems on the QTP. The review will also help bring to light the effects of grassland degradation on biodiversity–ecosystem function and design the way forward in maintaining the grassland ecosystem on the QTP.

*Methodology*

The data for this study was from secondary sources, such as peer-reviewed articles on the QTP and other grasslands, conference proceedings, newspapers, reports from various Chinese ministries' concern with the environment and agriculture, ecological restoration project reports, and personal observation in the field. In total, 146 articles and other source of materials were reviewed.

## 2. Status of Grassland Degradation on the QTP

Globally, the constant degradation of grasslands is a universal problem [33–35]. Grassland degradation, due to climate change and anthropogenic activities, has been regarded as one of the primary sustainable development barriers in the 21st century (Figure 1) [17,30,36,37]. Specifically, in China, a report by the National Development and Reform Commission of the People's Republic China

in 2015 estimated that approximately 22% of China's total land area is being degraded [38,39]. It is an undeniable fact with substantial shreds of evidence from the scientific community and local stockholders that the QTP grasslands have been experiencing degradation over the past decades [2,31,40]. A recent review by Cao et al. [34] noted those harsh environmental conditions, overgrazing, climate change, fragile soil, small mammals, privatization, and sedentarization are the principal factors causing the degradation of the QTP's grasslands. According to other reports, nearly two million km$^2$ of the grasslands on the QTP have been depleted, leading to a 30% reduction in productivity over the past two decades [3,27]. The health of grassland ecosystems can profoundly impact biodiversity, both directly and indirectly, due to the fact that native flora and fauna can cope with future developments of these environments [41]. Across other parts of the world, in Brazil, for example, the primary causes of grasslands degradation were the conversion of grassland into pastoral fields, farms, or afforested plantations [42]. In Pakistan, 79.1% of grassland degradation was due to anthropogenic activities, while 6.7% of restored grasslands were being impacted by climatic patterns. Similarly, in Mongolia, 85.1% of grasslands were influenced by climatic patterns, while 65% were disturbed by human activities. On the contrary, human actions accounted for 11.6% of grassland degradation in Uzbekistan [42].

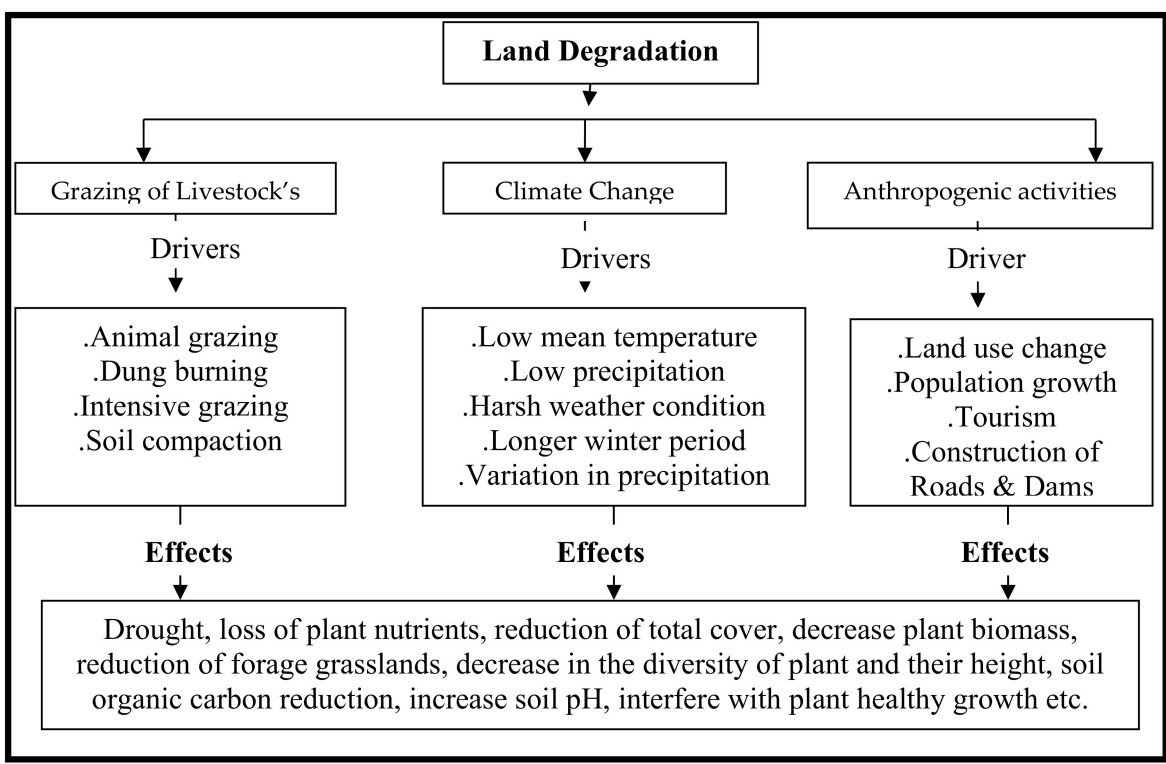

**Figure 1.** Key drivers and impacts of grassland degradation on the Qinghai Tibet Plateau.

*2.1. Quantification Attempts of Degraded Grasslands on the QTP*

The quantification of degraded grasslands across the QTP has been contentious with conflicting estimates by various authors and local authorities [41,43]. Reaching a consensus on the exact percentage of degraded grasslands has been a challenge in the past [34,41,44] and present. A couple of quantification methods have been employed to assess the rate of grassland degradation on the QTP appropriately, but the estimated values are inconsistent [45]. For example, quantification approaches like remote sensing [45], normalized difference vegetation index (NDVI) [46], and Landsat-8 satellite [32] methods have been used to detect degraded grasslands on the QTP over the past decades (Table 2). The results have been contradictory. Critics argue that until the principal causes of grassland degradation and trend are understood and categorized, estimating the exact degradation percentage will be impossible [41,47]. Currently, as the cause of degradation is scientifically unclear, the degradation estimates are inconsistent

and the restoration measures and strategies are not holistic in nature. The table below presents various attempted estimates of grassland degradation on the QTP (Table 2). The estimate approaches used in Table 2 were remote sensing, NDVI-LAI, (Leaf Area Index), Landsat satellites, and field inventory.

**Table 2.** Quantification estimates/assumptions of degraded grasslands on the Qinghai Tibet Plateau

| Ecosystem Type and Location | Degradation Estimates/Assumptions | Comments | Reference |
| --- | --- | --- | --- |
| Across QTP | 20–30% | Assumption | [47] |
| National level (China) | 90% | Assumption | [48] |
| Across QTP | 18.1% (1980) | FAO report | [49] |
| Across QTP | 28% (1990) | FAO report | [49] |
| Alpine grassland on QTP | 90% | Approximation | [50] |
| Black soil beach on QTP | 26% | Approximation | [50] |
| Alpine grassland QTP | 30% | Estimate | [51] |
| Alpine grassland on QTP | 90% | Assumption | [52] |
| Black soil beach on QTP | 35% | Estimate | [52] |
| Alpine meadow on QTP | 21% | Estimate | [53] |
| Across QTP | 57.19% (1996–2003) | Estimate | [54] |
| Across QTP | 19.55% (2003–2009) | Estimate | [54] |
| Across QTP | 40% | Assumption | [55] |
| Across QTP | 38.8% | Estimate | [42] |

## 2.2. Drivers of Grassland Degradation on the QTP

Both natural and human-induced drivers have continuously changed the structures and functions of grassland ecosystems from time to time on the QTP [2]. The sustainability of the QTP has been challenged by severe grassland degradation, mainly due to anthropogenic activities and climate change [3,42]. Besides climate change and anthropogenic activities, population increase and rodent damages have also been proven to contribute immensely to grassland degradation on the QTP [3,35,56]. According to Dingguo [1], the causes of grassland degradation on the QTP can be classified into the following categories: (1) inappropriate grassland use, (2) poor grassland management, (3) rodent and insect infestation, (4) forest clearance, and (5) medicinal herb collection and wasteland reclamation. Alternately, Liu et al., [3] listed glacial retreats, permafrost degradation, drying of wetlands, shrinking of lakes, and rodent root mass destruction as the environmental factors of grassland degradation. Additionally, Liu et al. [3] classified grassland degradation on the QTP into just two categories, namely, environmental factors and socioeconomic factors. Furthermore, infrastructural development measures undertaken across the QTP, such as railway construction, contribute to the degradation of the grasslands [57]. Other human factors, such as land cultivation and urbanization, have also been reported to contribute to land degradation [55].

Over the past century, grassland sizes of QTP have been shrinking, mainly due to population increase, climate change, rodent damage, and overgrazing [41]. Similarly, crop encroachment, urbanization [55], other anthropogenic activities [58–62], social practices, and ecosystem fragility [2] have led to grassland size reduction. The degradation of QTP started back from the 1980s, but became more eminent in the mid-1990s [7]. Population growth and the pressure to feed pastoral family members residing with pastoralists have led to the constant extension of arable lands, resulting in degradation [63]. Other factors such as land-use changes, shrub encroachment, droughts, and emigration [64] are also said to play a role in rangeland degradation. Grassland degradation is particularly noticed in southern Qinghai, Qaidam basin, and northern Tibetan landscape [7]. The strategic points of degradation are experienced on the plateau surfaces of smooth terrain, winter–spring lands, pedestrian routes, and banks of rivers [1].

## 2.3. Grassland Degradation as a Result of Anthropogenic Activities

Human interventions like urbanization, industrialization, and grassland cultivation coupled with wood and constant herb harvesting for fuel and production of Chinese traditional medicines have led to

the severe degradation of grasslands [3,65,66]. This severe degradation has led to enormous ecological consequences, ranging from a decline of plant species richness to the reduction of grazing lands [67,68] and food production [41,55]. The increase in population along the QTP, together with the demand for food, has exerted enormous pressure on grasslands, surpassing other factors such as climate change and damage done by rodents [63]. Population increase, high income, and the consumption demand for protein (meat) within China and out of China directly correlate to the massive pressure exerted on grasslands, hence causing their degradation. The presence of grazer-dwelling tents adjacent to the grasslands has proven to serve as a catalyst driving the degradation in many ways [12]. Liu et al. [63] found a significant correlation between degradation and settlement distance ranging from 4 to 12 km on the QTP. Infrastructural development like railways, hydro-electrical poles, roads, and bridges connecting other parts of the QTP have equally contributed to the degradation of the grassland. The use of big and heavy machineries during road and railway construction damages biodiversity and leads to soil nutrient loss and compaction [7].

## 2.4. Degradation Due to Intensive Overgrazing

By 2005, it was estimated that the QTP's grasslands supported approximately 12 million yaks and 30 million sheep and goats, which are far beyond the carrying capacity of the grassland ecosystems [41]. Overgrazing by livestock has been recorded to degrade grasslands by accelerating soil erosions [69]. It also serves as a critical factor that influences productivity, vegetation structure, and grassland nutrition [70]; stimulates landscape fragmentation [71]; deteriorates grassland vegetation [22,57]; and induces grassland degradation [34]. The design of effective management strategies for livestock grazing may require an in-depth understanding of the impact of grazing across the QTP [33]. Overgrazing on the QTP is mainly caused by Yaks and sheep (Figures 2 and 3).

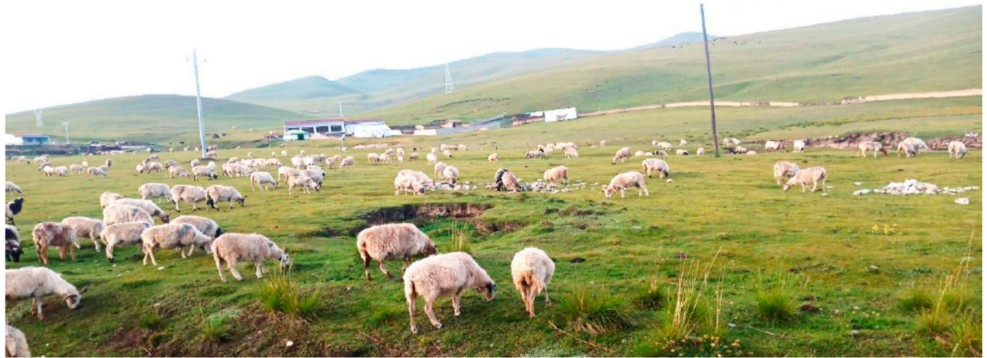

**Figure 2.** Sheep grazing on the Qinghai–Tibet Plateau

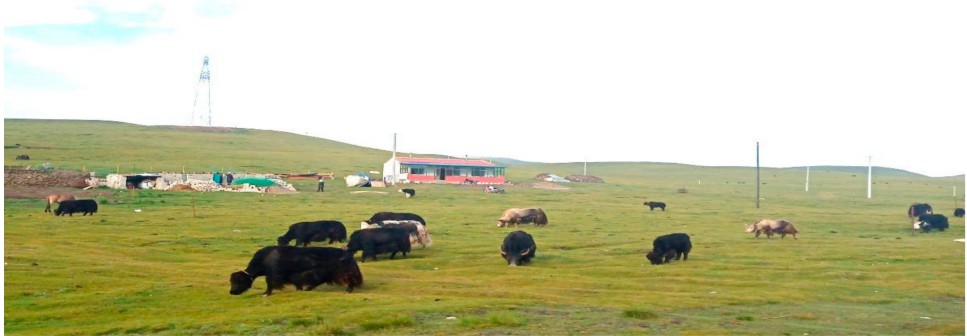

**Figure 3.** Yak grazing on the Qinghai–Tibet Plateau.

### 2.5. Degradation Caused by Climate Change

The QTP is commonly called the "roof of the Earth" due to its high elevation, making it closer to harsh and extreme weather conditions [72–75] and degradation [69]. Climate change is considered to be one of the main causes of grassland degradation across the QTP [42,57]. Literature has proven that climate change accounts for 56.7% of grassland degradation across the QTP [42]. The impact of climate change on the QTP has been felt in many ways, such as desert-extreme degradation, species extinction, decrease in biodiversity, and constant plant cover decrease over the years [25,41,55,72,73]. The unpredictable climate patterns have also influenced the river water quantity, storage, and flows, thereby affecting communities relying on water from the upper QTP region [55]. This pattern is also affecting species' ability to cope with harsh climatic conditions that are causing the deterioration of the grassland ecosystems [74] over time. According to recent scientific findings, it has been proven that the QTP warming is more severe than the global temperature on the average [75,76]. This warming mainly occurs from South China to North China with a similar temperature. Northern China experiences the most significant warming [77] compared to South China. From 1984 to 2009, the warming rate was 0.46 °C per decade, which exceeded that of the Northern Hemisphere (0.32 °C per decade). This warming rate was 1.5 times higher when compared to the global warming trend [75,77]. Li et al. [69] concluded that climatic factors stimulating grassland dynamism are spatially heterogeneous [69]. The altitude of the QTP must have contributed to the growth of specific species tolerant to high elevation. Most species that grow in adjacent communities might not be able to cope with the 4000 m elevation above ground level.

### 2.6. Degradation Caused by Small Rodents

Recent studies by Dong et al. [2] affirmed that rodent damage is the key cause and stimulator of grassland degradation on the QTP. *Ochotona curzoniae*, commonly called pikas, is the dominant rodent type found on the plateau (Figure 4). Arthur et al. [78] counted approximately 2183–4423 rodent holes per ha in the southern part of the QTB and concluded that 10% of the grassland plant biomass of the plateau was consumed by pikas. The increase in rodent population has led to grassland degradation through their burrowing and grass consumption activities [2,79]. Scholars like Zhang et al. [80], Jones et al. [81], and Brown and Heske [82] argued that smaller mammals, such as pikas, are beneficial to the ecosystem by facilitating nutrient cycling and direct physical effects.

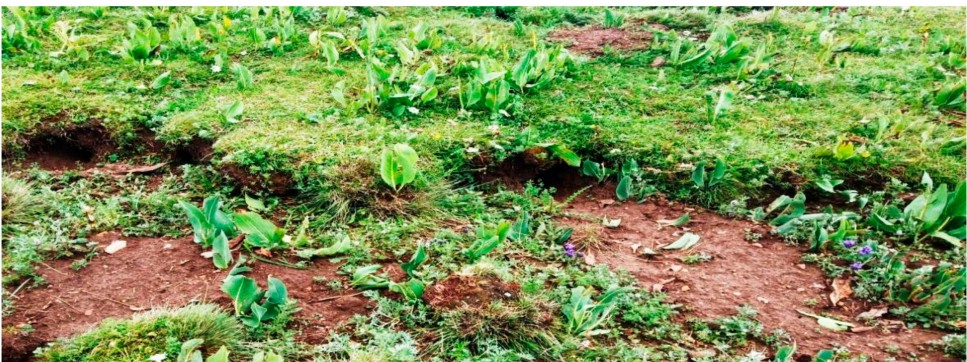

**Figure 4.** *Cont.*

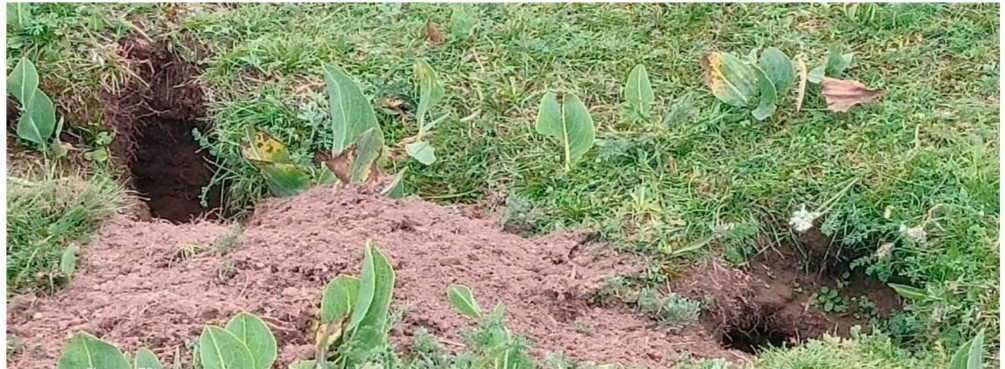

**Figure 4.** Burrowing holes of pika and zoko (photos taken in Maqin county).

*2.7. Other Uncertain Factors of Grassland Degradation on the QTP*

Other causes of grassland degradation may include various developmental activities, such as highway and railway construction. For instance, the highways along the QTP had increased by 3.6-fold from 2000 to 2019. Infrastructural development projects like highway construction can lead to ecosystem fragmentation, water and air pollution, and blocking of waterways [7]. Besides this, improper reclamation approaches have also been reported to contribute to land degradation on the plateau [27]. Tourism and influx of natives from other regions have also been considered as a grassland degradation factor [83]. Le et al. [83] concluded that tourism exerted adverse effects on the grass by lowering the plant height, species diversity, and above-ground biomass leading to soil degradation as a result of decrease in chemical properties. Decoupling human–nature systems and ploughing of grasslands for crop cultivation have also contributed to the degradation of grassland ecology on the QTP [2,3,13,27,41,84]. Similarly, erosion due to traditional pastoral management approaches adopted by livestock grazers on the QTP also contributes to grassland degradation [34].

*2.8. Grassland Degradation Versus Livelihood Challenge on the QTP*

Pastoralism has been the critical source of livelihood on the QTP for centuries. Livestock grazers have been receiving numerous benefits from this profession in the form of wool, milk, meat, income, and fuel to meet their domestic needs on a daily, monthly, and yearly basis [85]. Contrastingly, revenue realized by the sale of meats and milk is believed to outweigh the economic loss so far. The poverty scale among the QTP is estimated to be 36% [78] and sheep or yak farming is the sole source of survival for most families involved in livestock grazing. The residents of the QTP are solely dependent on pastoralism to meet their daily needs, and pastoralism is mostly done through livestock rearing. Large-scale grazing, such as the one being practiced on the QTP, not only fulfills family demands, but also contributes to national food security and economic growth through the exportation of meat and milk coming from the plateau. The degradation of grassland through grazing is not intentional, but a source of survival and culture for native Tibetans as their whole life is dependent on these practices. Interestingly, although livestock grazing is believed to be the critical livelihood source on the QTP, for some Tibetans, it is a way of life and cultural heritage worth being sustained and maintained for generations yet unborn.

## 3. Effects of Grassland Degradation on Biodiversity and Soil Properties on the Qinghai–Tibet Plateau

*3.1. Effects of Grassland Degradation on Plant Species Vegetation*

The degradation of grasslands on the plateau has caused a decrease in plant diversity, height, cover, and productivity [3,85]. Over the years, a plethora of scientific research has been carried out to assess the effects of grassland degradation on biodiversity–ecosystem function [28,29,33,86–88]. Recent studies conducted on the QTP discovered that plant biomass has been steadily decreasing [29,89] and forage

grasslands have also reduced drastically [14]. According to findings from Li et al. [7], the above-ground biomass has decreased by 4–16 kg ha$^{-1}$ yr$^{-1}$ on the plateau. Grassland degradation has reduced plant speciation and have affected the distribution of plant species across the plateau [90]. The degradation of grasslands within the QTP has not only threatened the biodiversity function potential, but also plant species distribution, composition, and diversity [33]. Wang et al. [84] reported that degradation decreased the plant and total biomass of Cyperaceae, Poaceae, and Fabaceae in all plant communities. Additionally, grassland degradation has led to severe deterioration in biodiversity–ecosystem function and services such as soil aeration, water holding capacity, carbon accumulation, and plant biomass storage [28,54]. Furthermore, grassland degradation has resulted in soil organic carbon loss [91]; reduced plant cover and ecosystem carbon and nitrogen storage [92]; negatively affected the ecological security and local economy of China [42]; transformed grasslands into a harsh environment [69]; caused a decline in soil organic matter [93]; reduced traditional biomass by locals [67]; reduced soil moisture content due to disturbance by pikas on the plateau; and caused a decline in soil organic carbon [56].

## 3.2. Effects of Grassland Degradation on Soil Properties

Recent studies have proven that grassland degradation leads to more significant impacts on soil physical and chemical properties [2,27,29,53,80]. Lu et al. [33] concluded that soil organic carbon, total soil nitrogen, and microbial biomass carbon were reduced significantly by overgrazing and other degradation drivers. Su et al. [50] stated that the soils tend to release carbon and nitrogen as poisonous greenhouse gases (e.g., $CO_2$, $CH_4$, and $N_2O$) into the atmosphere when soils are severely degraded to a certain level. The gradual decrease in plant cover leads to many other issues, such as increased erosion, increase in pH, and nutrient loss on soil surface. The degradation of grasslands on the QTP has not only had effects on the vegetation performance and distribution, but also on soil nutrient availability uptake. It has been reported that 42% ± 2% of soil organic carbon (SOC) has been lost on the QTP as a result of grassland degradation from various factors. Furthermore, land-use changes, extreme weather conditions, and intensive overgrazing have influenced the amount of nutrients stored in grasslands on the QTP. According to Dong et al. [2], vegetation and soil nutrient imbalance is also considered as an essential factor in grassland degradation on the plateau. It was reported by different case studies that grassland degradation has reduced total nitrogen by 33% ± 6% [3], total phosphorus by 17% ± 4% [94], and potassium by 15% ± 3% [61] on degraded grasslands. There is a correlation between grassland ecosystem health and the amount of nutrient the soil can store; hence, the decrease in grassland cover will definitely affect soil properties. Grassland degradation on the QTP has resulted in soil nutrient decline and variation across grassland types (Table 3).

**Table 3.** Variations in soil organic matter and total nitrogen of alpine soils [11].

| Soil Type | OM (%) | Total N (%) | Sample No |
|---|---|---|---|
| Alpine meadow soil | 10.7 | 0.47 | 11 |
| Subalpine meadow soil | 15.7 | 0.69 | 13 |
| Alpine steppe soil | 1.7 | 0.12 | 6 |
| Subalpine steppe soil | 3.1 | 0.20 | 8 |
| Alpine desert soil | 0.49 | 0.04 | 2 |
| Subalpine desert soil | 0.76 | 0.06 | 2 |
| Alpine frigid soil | 0.79 | 0.06 | 7 |

Moreover, grassland degradation has led to leaching and an increase in wind and dust frequency and pollutes in both surface water and ground water [3], and has stimulated the erosion of sand and soil silt into rivers [27]. A summary of grassland degradation drivers and their adverse effects is presented in Table 4. A combination of major grassland degradation drivers, such as human activities, climate change, overgrazing, rodent burrowing, and other minor degradation drivers, has stimulated the decline in soil nutrients, plant biomass, and biodiversity across the QTP (Table 4).

**Table 4.** Summary of grassland degradation drivers on the QTP and their adverse effects.

| Degradation Drivers of QTP Grasslands | Negative Effects | Sources |
|---|---|---|
| 1. Anthropogenic activities | Ploughing grasslands for crop cultivation resulted in extensive grassland degradation, stimulated desert exacerbation over time, reduced the carbon cycle of terrestrial ecosystem, and induced alpine ecosystem change. Road and railway construction affected the vigilance behavior and initial flight ability of wild birds, and decreased aboveground net primary productivity (ANPP). | [2,85,95–98] |
| 2. Climate change | Increased unstable plant biomass; decreased plant cover and aboveground and belowground biomass; stimulated desert exacerbation; declined river water quantity, storage, and flows; decreased grassland quality and species richness, limiting precipitation and temperature; reduced permafrost and glacier receding; altered soil carbon and nitrogen cycling; and transformed alpine meadow into shrubs. | [25,34,41,55,60,72,73,85,99–101] |
| 3. Grazing | Altered the surface of the grassland physical environment, changed the belowground biomass, accelerated soil erosion and the loss of soil nutrients, increased landscape fragmentation, altered the plant life form as well as the plant population, decreased the plant species abundance, altered the composition and structure of plant communities, decreased soil moisture, and negatively influenced grassland vegetation. | [2,22,69–71,102–105] |
| 4. Burrowing activities of rodents; pikas (*Ochotona curzoniae*) and zoko (*Eospalax fontanierii*) | Decreased biomass productivity, stimulated the expansion of bare patches, damaged alpine meadow vegetation, declined ecosystem production, lowered plant cover and soil nutrient plant productivity, and reduced grassland ecosystem functions and services. | [34,106–112] |
| 5. Other activities | The harsh environment and natural disasters stimulated the decline in plant cover and the conversion of rangelands into agricultural lands. Archaic livestock husbandry approach and privatization have also contributed to the decline in vegetation cover, reduction in plant productivity, and acceleration of topsoil erosion. Downward drainage of water resulted in the drying of topsoil and permafrost decline reduced the activities of soil microbes. | [34,41,42,60,84,113–115] |

## 4. Grassland Ecosystem Restoration and Rehabilitation Efforts: Improved Grassland Management Practices

### 4.1. Grassland Ecosystem Restoration Efforts at a Global Scale

The advancement in knowledge and technological innovations can be observed in the form of environmental restoration techniques and approaches that have been recently adopted worldwide. For example, the use of remote sensing in ecological ecosystem management has paved the way for land managers, practitioners, and policymakers to measure ecological ecosystem losses and gains at multiple spatial and temporal scales [116]. Remote sensing technology has the ability to assist land developers and other practitioners to locate potential areas suitable for restoration and to identify possible sound restoration objectives and institute them, while monitoring their progress [116]. In fact, in recent times, remote sensing tools and ideas have been used widely by many scholars [117–120] to promote sustainable restoration and provide insight to policymakers. Furthermore, inventories of land coverage derived from remote sensing and aerial photography are comparatively straightforward and can produce distinctions primarily based on obvious broad classes in powerfully influenced habitats [121]. Besides remote sensing, other aerial technologies, such as the Landsat-8 and normalized difference vegetation index (NDVI), are helpful in ecosystem restoration.

Traditional and inexpensive popular restoration methods adopted in the past and in recent times in ecological restoration include re-vegetation [122], habitat enhancement, remediation, appropriate mitigation [123], biophysicochemical methods for rivers affected by pollution, fencing of grasslands, afforestation, and sand protection using nylon meshes for desert ecosystem [108]. These conventional field methods are expensive to undertake, and the results are often unreliable [48] due to large grassland areas. Recently, a new concept put forward by Butterfield et al. [124] suggests that pre-restoration approach involved in restoring species that are suitable in a particular site, now and in the long run, are perfect mitigation measures. They propose a technique for taxonomic classification of new species that will appropriately reward long-term losses of appropriate sites by present target species [124]. For example, the Colorado Plateau, alongside other states and agencies in the United States, has already developed 26 grass species with great restoration potential for diverse ecological habitats [125].

### 4.2. Chinese Authority's Interventions in Combating Grassland Degradation and Ensuring Ecological Restoration Across Grasslands and other Ecosystems in China

The Chinese government has embarked on several ecological restoration projects since the 1970s to mitigate the rate at which grasslands and other ecological systems are dilapidated across China [35,55]. In total, six mega ecological restoration projects have been undertaken in China over the past three or more decades and covered approximately 23.2% of the grassland areas of China [35]. Ecological restoration projects are critical and vital approaches necessary to stimulate ecosystem resilience to withstand and respond to environmental dilapidation and other human interventions [55]. Projects such as "*Returning Grazing Land to Grassland Project 2003–2010*" were aimed at the ecological restoration of grazed ecosystems. The "*Grain for Green Program 2001–2010*" was aimed at ecological restoration and compensation. The "*Restoring Grasslands of the Qinghai–Tibet Plateau 2017–2019*" project was designed to halt grassland degradation on the QTP. The "*Beijing–Tianjin Sand Source Control Project 2001–2010*" was tailored to promote environmental conservation through sand dune control within the suburb of the Beijing city. The "*River Shelter Forest*" was the abbreviation for the title of the "*Yangtze River and Zhujiang River Shelter Forest Project*" that was established in 1989 and completed in 2010. The ecological mandate of this project was to halt flooding and regulate soil erosion in South China [3]. The outcome of these initiatives resulted in the establishment of 214 critical approaches and technologies, over 100 advance technological systems, and 64 ecological restoration models [126,127]. A recent study conducted by Lu et al. [35] concluded that the six mega ecological projects implemented across China contributed positively to carbon sequestration and plant cover, and reduced soil erosion across various

ecosystems in China. A few of the restoration work started in the early 2000s, both by the government and scientific community, in collaboration with the local stakeholders.

At the local level, an initiative titled "*Retire Livestock and Restore Pastures*" was formed in 2003 to regulate grazing activities on the QTP [45,113]. This initiate incorporates ecological engineering concept and model to halt the degradation of grasslands through rotational grazing and exclusion approaches. In the Menyuan Hui autonomous county, Qinghai province, an "*Eco-restoration*" project was also initiated by the local county authorities to ease the enormous grazing pressure on the QTP [102]. Additionally, Dong et al. [80,94] highlighted five grassland restoration methods employed across the Qinghai–Tibet Plateau. These approaches are bare land re-vegetation methods that use cultivated grasslands and include: (1) revegetation of bare land with cultivated grasslands; (2) grass combination and mix sowing; (3) selection of native plant species for restoration; (4) irrigation and fertilization; and (5) ecological replacement. Other approaches have equally been employed to find a suitable and amicable means of combating grassland degradation on the QTP, although their outcomes brought mix reactions and uncertainties [41]. Approaches that use, for example, fertilization, weeding, rodenticides [128], reseeding, solar energy use, and biogas usage [63] have already been employed for grassland restoration on the QTP. Su et al. [50] concluded that re-vegetation can help restore the total carbon and nitrogen in the soil of alpine grasslands on the QTP. Although it can be a cumbersome, time-consuming, and expensive process, if undertaken, it can have a positive impact on grasslands on the QTP. Another possible measure worth noting in the fight against grassland degradation is grazing exclusion. Wang et al. [105] and Liu et al. [3] suggested that grazing exclusion could be another effective management plan to restore degraded grasslands. This strategy could, however, reduce grazing lands for cattle, thereby reducing the income for pastoralists and all who depend on livestock rearing for survival. Nonetheless, if exclusive grazing is well planned, it can yield better results. Measures such as weeding, rodenticide use, and fertilization have also been applied to reduce grassland degradation on the QTP [128], but their outcomes were not satisfactory. Zheng et al. [126] listed five restoration approaches being adopted by local herders within the QTP, i.e., grazing enclosure, forage crop (annual herbs) cultivation, grazing prohibition, derivatization, and grass (perennial herbs) seeding. Zheng et al. [126] concluded that these locally adopted approaches have proven to be efficient in some locations and farmers are pleased with the outcome, indicating that balance in ecological conservation and economic development is conceivable if sound management measures are adopted. Liu et al. [3] cautioned that any attempt to cultivate crops only suited for livestock consumption might also lead to grassland degradation. These projects helped transform severely degraded grasslands into beautiful, scenic, flowering landscapes that attract tourists and other visitors to the county. A study conducted by Cai et al. [55] found that ecological restoration projects implemented on the QTP helped to reverse and mitigate the degradation of the grasslands and increase net primary production and vegetation cover in many areas.

The grassland degradation problem is still a complex challenge faced by ecologists and other environmentalists in China and beyond. In order to understand the problem of grassland degradation, the causes must be understood first (Figure 5). Understanding the critical drivers of grassland degradation on the QTP can help in designing sound mitigation measures to be adopted across different grassland types. Secondly, when the grassland degradation drivers are known, an expert's opinion and services can be easily sought. The knowledge of key degradation drivers can also help in identifying the extent of areas to be quantified and the appropriate methods that can be employed. Having the required expertise and knowledge on the extent of degradation can help policymakers and the scientific community to make informed decisions on long- and short-term mitigation strategies for QTP grasslands. A conceptual framework illustrated in Figure 5 describes possible actions needed to be undertaken before a suitable solution on grassland degradation can be achieved. A combination of good policy instruments and accurate grassland degradation percentage estimates can help in putting together sustainable management approaches that will ensure ecological stability and grazer-friendly plans (Figure 5).

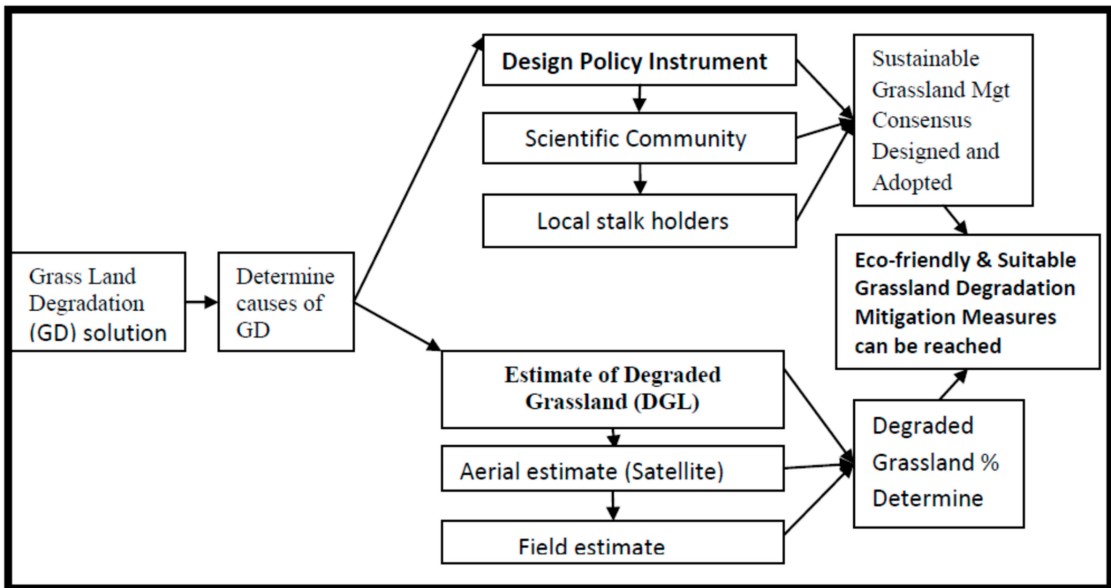

**Figure 5.** A conceptual framework to mitigate grassland degradation

*4.3. Grassland Ecosystem Restoration Outcomes on the QTP*

The biodiversity–ecosystem function can be referred to as various biological processes of the ecosystem and allied services, such as the provision of clean air, water, and food [127–134]. Although restoring degraded grassland ecosystems are complex and often involve huge money [48], conservation, as well as restoration, has been proven to have positive outcomes [26,130,133]. Similarly, landscape modification through ecological restoration has both positive and negative influences on human livelihood [135]. For example, the cultural aspect of the ecosystem when restored provides aesthetic services [136], supports interaction and benefits from the habitat [137,138], promotes recreational services and facilities [139], fulfills cultural heritage and desires, [140], and supports spiritual obligations and worship [141]. The positive results of most restoration projects are felt and seen at the local level before being realized at a national or international scale [142–146]. On the other hand, ecological restoration in some cases affects and changes the way of life and the livelihood pattern of affected and adjacent communities [56]. Additionally, Huang et al. [28] noted that payment for ecosystem services improves ecological conservation and restoration at a local scale.

## 5. Conclusions and the Way Forward

Climate change and anthropogenic activities have been the principal causes for grassland degradation on the QTP. Other minor drivers include rodent damage, urbanization, land reclamation, construction of highways and railways, agricultural cultivation, tourism, etc. The degradation of grasslands has produced adverse effects on biodiversity and soil potential on the QTP and its immediate environment. Plant productivity, diversity, and cover have declined due to ongoing grassland degradation on the plateau. Similarly, soil organic carbon, total nitrogen, and total phosphorus have declined steadily. The provision of a suitable alternative livelihood within the QTP is the first and most important criteria to embark on, if land degradation is to be mitigated and managed in a sustainable manner. The provision of lucrative alternative livelihood at the doorstep of local people living on the QTP can cushion the degradation problem. An alternative livelihood scheme should be a need for the local communities rather than an advanced option; otherwise, the people would easily become disinterested in the scheme and its adoption will be abandoned before implementation. Where suitable and flexible alternatives are provided, this will stimulate natives to adopt such programs and hence minimize their dependence on livestock grazing.

The inconsistent grassland degradation estimates put forward by different authors are proving difficult to be used to comprehend the nature of degradation, thus, posing a challenge in tailoring policy instruments that will help mitigate the degradation problem. A holistic and national survey is urgently needed to assess and demarcate the degraded grasslands on the QTP on a regular basis, so that the scientific community can have consistent data. The sustainable management of QTP's grasslands requires adequate knowledge about the long-term environmental consequences of grazing, root causes of grassland degradation, and the strategies and approaches needed to mitigate degradation without destroying the livelihood and survival privileges of local communities. Grassland rotation needs to be promoted to mitigate grassland degradation on the QTP in the near future. The rotation of grasslands entails a systematic and sequential shuffling of browsing areas every 3 or 6 months. This approach could encourage fresh grass to grow in one section, while the other section can be utilized by livestock. This approach is similar to the ex-closure method that is already being adopted by some pastoralists on the QTP; however, this method has a shorter rotation length compared to ex-closure that can take years. The ex-closure approach that is currently being practiced has shown that grassland degradation can be controlled if there is willingness from local communities. This approach not only supports systematic grazing, but also enhances research to compare the degradation levels across various regimes. Too large an ex-closure area will make the grazing land unavailable for the growing number of livestock.

The adoption and distribution of irrigation channels across the QTP could be another option to mitigate grassland degradation as a result of prolonged drought periods. Since summer or spring season is short on the QTP, substitution with irrigation could be ideal to ensure the availability of grasses around the clock. The use of N and P fertilizers may also yield better results as they help the plants to grow fast in a short time. Similarly, the adoption of appropriate measures to eradicate poisonous plant species as a result of overgrazing could be helpful for the healthy growth of grasses on the QTP. Forage crops can be planted to compete with native forage and to support livestock during grass-deficit periods. A sustainable management strategy is needed to combat the extinction and exploitation of biodiversity across the QTP's grasslands. A policy consensus between government authorities and traditional grazers on strategies to sustainably manage the grasslands should also be developed or updated, if available, and should follow a win-win approach for both parties.

**Author Contributions:** M.F. conceived and designed the research; S.D. edited, supervised and directed the article layout; S.W.K. helped to design, advice, and proofread the article; S.A.U.R. edited and provided useful advice and resources for the article; M.Y. and J.X. helped in gathering necessary information needed to write the article. The above authors contributed meaningfully to manuscript development and gave their full approval for publication submission. All authors have read and agreed to the published version of the manuscript.

**Funding:** This research was financially supported by the grants from the Second Tibetan Plateau Scientific Expedition and Research Program (2019QZKK0307), National Key R&D Program of China (2016YFC0501906), Qinghai Provincial Key R&D program (2019-SF-145 and 2018-NK-A2), and Qinghai innovation platform construction project (2017-ZJ-Y20). The authors would also like to thank the anonymous reviewers for their helpful comments.

**Acknowledgments:** A special thanks is extended to the Tiebujia, Maqin and Maduo grassland management and local authorities for their support during a visit for this research.

**Conflicts of Interest:** The authors declare no conflicts of interest.

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
