# Peer review of "Status and Challenges of Qinghai–Tibet Plateau’s Grasslands: An Analysis of Causes, Mitigation Measures, and Way Forward"

_sustainability, doi:10.3390/su12031099_

Round 1
Reviewer 1 Report
This article is quite interesting and easy to follow.
The English just needs a minor review asnd revision. There is not much to be revised.
Please make sure that all the tables are referred to in the text (this is done for the Tables) and the Figures including the photos should also be referred to in the text.
Otherwise the article is interesting and makes a good contribution to the domain.
Reviewer 2 Report
This review of the literature related Qinghai-Tibet Plateau grassland degradation amounts, causes and mitigation projects is significant. The paper tries to use this to identify possible policy and design solutions to combat and/or reverse the types of ecosystem damages identified.
The studies cited appear to be relevant and recent, but could benefit from more diversity of research to include lessons learnt from other parts of the world. Some of the in-line citation numbers appear to be incorrect (eg, [63] on Line 179). Reference list format needs editing as well.
Since this is a review paper, synthesis and interpretation are important. Unfortunately, this paper does not do a very good job of either. I believe the problem may be with the language and re-editing for language and syntax may improve things. Here are a few specific examples
Paper is too repetitive - eg, Table 1 is referenced in sections 1.0 and 2.0 in the exact same way. Terms should be explained more. Eg Line 138: Restoration Rate. Line 138-139: Reference needed for restoration rate figure. Line 149-150: Reference needed for the critiques statement. Also, why should not knowing the cause of degradation hinder quantifying amount of degradation? Table 2 may benefit from including estimation method used in each instance (NDVI, LANDSAT, field surveys etc) Line 215-216: Why refer to grazing when discussing effects of climate change? Section 2.5 needs significant editing to improve clarity and readability Section 2.5: Grazing caused 42% grassland degradation (Line 216) but human activities only caused 19.9% grassland degredation (Line 224)? Line 226-227: If South to North warming is uniform in temperature, how does the North face most significant warming? South to North what… hemisphere? South China to North China?
Section 2.3 Grassland degradation as a result of anthropogenic activities does not discuss all the anthropogenic drivers listed in Figure 1, especially infrastructure development. This is a very significant driver to leave out of the discussion. It seems that dam and road building is identified as a driver of land degradation and yet, irrigation canals are proposed as a viable solution to the problem (Line 457-459).
In conclusion, this is an important paper as it collates the state of knowledge about this subject, but it needs significant improvements in language and synthesis of the studies cited.
Reviewer 3 Report
It would be good to describe more detail
Figure 5. Conceptual framework to mitigate grassland degradation,
because it is the key result of the paper!
Reviewer 4 Report
Dear authors,
In my opinion the paper fits the scope of the Sustainability well. I also believe the paper can make a significant contribution to the literature. For this reason, I will recommend its publication if you do these major changes:
*Keywords:
There are some unnecessary key words as “ecosystem”. “Qinghai-Tibet Plateau” and “Grassland” are included in the tittle, so is not necessary toad it as key words. I recommend delete it.
*Introduction:
The Introduction should be more descriptive to the relevance of this study. Actually, it presents a very short description regarding the study area. I can’t find the relevance of this research in this section. I suggest linking section 1.2. into Introduction section.
The objective of the research is not clear. I propose you define it clearly in this section.
*Moreover, the assessed items should be justified with literature in a section related to literature background or similar.
*Methodology:
I miss a methodology section, where the methods that should be explained. I suggest explaining the source or the origin of the data and the nature of the data in this section.
*Table 1 is no complete. Please, solve it.
*Table 4: review the source and style, please.
*Figure 5: idem.

Round 2
Reviewer 2 Report
The authors have addressed reviewer's comment.
Author Response
The author's have addressed reviewer's comments.
Reviewer 4 Report
Dear authors,
Thank you to improve the manuscript. I suggest specify the number of sources reviewed and change the numeration of section 1.1. Methodology. It's the same that section 1.1. Introduction.
Kind regards
